# Extracellular Vesicles Derived from Mesenchymal Stem Cells Promote Wound Healing and Skin Regeneration by Modulating Multiple Cellular Changes: A Brief Review

**DOI:** 10.3390/genes14081516

**Published:** 2023-07-25

**Authors:** Weiyuan Zhang, Yang Ling, Yang Sun, Fengjun Xiao, Lisheng Wang

**Affiliations:** 1Department of Special Medicine, School of Basic Medicine, Qingdao University, Qingdao 266000, China; zhangwy_114@163.com (W.Z.); lingyang0532@163.com (Y.L.); suny201966@163.com (Y.S.); 2Beijing Institute of Radiation Medicine, Beijing 100850, China

**Keywords:** mesenchymal stem cells, cellular changes, extracellular vesicles, wound healing, skin regeneration

## Abstract

Mesenchymal stem cell-derived extracellular vesicles (MSC-EVs) are biologically active substances secreted by MSCs into the extracellular matrix that play an immunomodulatory role in skin damage repair. To investigate the mechanism of MSC-EVs in reducing inflammation, promoting angiogenesis, promoting the proliferation and migration of epithelial cells and fibroblasts, and extracellular matrix remodeling during wound healing, we focused on the effects of EVs on multiple cell types at various stages of skin injury. A literature review was conducted to explore related research on the influence of MSC-EVs on the types of cells involved in wound healing. MSC-EVs show a strong regulatory ability on immune cells involved in the regulation of inflammation, including macrophages, neutrophils, and T cells, and other cells involved in tissue proliferation and remodeling, such as fibroblasts, keratinocytes, and endothelial cells, during wound healing in in vitro and in vivo experiments, which substantially promoted the understanding of wound healing in the field of trauma medicine. MSC-EVs have potential applications in combating poor skin wound healing. Elucidating the mechanism of action of EVs in the wound-healing process would greatly advance the understanding of therapeutic wound healing.

## 1. Introduction

Poor skin wound healing is an urgent concern in the field of trauma medicine. The application of traditional therapeutic methods, such as systemic anti-inflammatory drugs and traditional dressings, has not achieved breakthroughs in skin wound healing. Mesenchymal stem cells (MSCs) were found to possess great therapeutic potential for wound healing and skin regeneration [1]. MSCs, also known as mesenchymal stromal cells, are recognized as cell populations with diverse differentiation potential and are derived from fat, umbilical cord, amniotic fluid, placenta, skin, dental pulp, and many other tissues [2]. MSCs were initially reported by Friedenstein et al. for their ability to self-renew and undergo multilineal differentiation [3,4]. Subsequently, researchers found that MSCs secrete various small molecules, such as extracellular vesicles (EVs), cytokines, chemokines, growth factors, and interleukins (ILs), which can undergo endocytosis or bind to receptor surface proteins, transmit signals to the corresponding receptor cells, and mediate intercellular communication among cell types to change their biological behavior and participate in immune regulation [5,6,7,8,9].

Researchers reported that the paracrine function of MSCs enables them to acquire strong immune regulation capabilities and have attempted to apply them to cell therapy regimens for various human diseases [10]. Some effects, such as suppression of the local immune system, inhibition of fibrosis (scarring) and apoptosis, enhancement of angiogenesis, stimulation of mitosis, induction of tissue intrinsic repair cells, and stem cell differentiation, are different from those of MSCs that differentiate directly into repair tissue [11]. MSC-EVs have been shown to promote skin wound healing and accelerate this process through multiple mechanisms. These mechanisms comprise reducing inflammation, promoting angiogenesis, and promoting proliferation and migration of epithelial cells and fibroblasts. Consequently, MSC-EVs can be used as new biomarkers and therapeutic targets because of the functional molecules they encapsulate, which can simultaneously promote wound healing through multiple mechanisms and may be a promising method to replace cells for skin wound treatment [8].

Researchers have explored the role of MSC-EVs in various ischemic tissue-damaging diseases, including skin wound healing, vascular remodeling, hair growth, and skin beauty. This article briefly introduces the development history of MSC-EVs, reviews the mechanisms of MSC-EVs in promoting wound healing and skin regeneration, summarizes the potential of MSC-EVs in promoting skin wound healing, and focuses on recent studies.

## 2. Description of MSC-EVs

MSCs are often described as a highly heterogeneous population of stem and progenitor cells that expand into unisolated fibroids and mucinous cells in vitro [12]. Initially, a group of fibroblast-like cells, capable of differentiating into adipocytes, chondrocytes, and osteocytes, was isolated from the bone marrow of guinea pigs and mice, which influenced the microenvironment for the in vitro culture of hematopoietic stem cells (HSCs) [3,4]. These cells were later identified as MSCs in human tissues. In 2006, the International Society for Cell & Gene Therapy provided a clear definition for MSCs. They express surface molecules, including CD105, CD73, and CD90 but do not express surface molecules, including CD45, CD34, CD14 or CD11b, CD79α or CD19, and HLA-DR, and can differentiate into osteoblasts, adipocytes, and chondroblasts in vitro [13]. More importantly, MSCs were later found to produce some “factors” through paracrine actions, which can play notable roles in immune regulation [14,15].

### 2.1. Classification, Labeling, Formation, and Delivery of MSC-EVs

MSC-EVs are a diverse family of particles composed of membrane-bound particles released from stem cells. Particles in this family are predominantly circular, isolated, or rarely aggregated into small clusters, have confinement membranes, and exhibit uniform electron transmission [16]. There are many types of MSC-EVs, and research scholars have followed two rules to classify them. The first rule is from the International Society of Extracellular Vesicles, which revised a new MSC-EV subtype nomenclature based on the physical properties, biochemical components, and cells of origin of EVs according to the latest information on EV research in 2018 [17]. Another method is simpler than that aforementioned and is a general classification according to the subtype of MSC-EVs, namely, exosomes (30–120 nm), microvesicles (MVs) (100–1000 nm), and apoptotic bodies (800–5000 nm) [18]. Markers of EVs include multiple proteins involved in endosome biogenesis, such as Alix, tumor susceptibility gene 101 protein (TSG101), tetraspanins (CD63, CD81, CD9), and lysosome-associated membrane proteins (LAMP1 and LAMP2) [19]. The membrane of MSC-EVs contains large amounts of cholesterol, sphingomyelin, ceramide, and various lipid molecules [20]. Additionally, the membrane surface of MSC-EVs was confirmed to contain both the characteristic surface markers of MSCs (CD29, CD105, and CD73) and the traditional markers of EVs (CD63, CD81, and CD9) [21]. EVs contain proteins, miRNAs, and lipids [22]. Regarding the formation of EVs, the budding theory, which refers to the formation of multivesicular bodies (MVBs) that fuse with the plasma membrane after mature endosomes sprout inward, is widely recognized. The buds released later are called EVs [8]. MSCs produce active EVs and release them into the cytoplasm of recipient cells, where they are captured by recipient cells through endocytosis, receptor–ligand binding, or direct binding and can transmit signals to recipient cells, guiding their biological behavior [23]. Under some conditions, small EVs were not easily distinguishable from exosomes, and some subpopulations of small EVs were similar in size to exosomes and were observed during direct budding from the plasma membrane [24]. MSC-EVs consist of many different molecules, such as nucleic acids (DNA, RNA, mRNA, and miRNA), pro-inflammatory and anti-inflammatory cytokines, enzymes, and various other proteins [24]. Some research scholars believe that MSC-EVs can not only dump self-secreted cytokines into the intercellular space for recognition by any cell containing the corresponding receptor but also deliver a small amount of cytokines directly to target cells, which is a surprising and more efficient delivery mechanism than that of MSCs [25].

### 2.2. MSCs and MSC-EVs

MSC-EVs may contain MSC-specific components and exert specific effects on recipient cells, similar to the therapeutic effect of MSCs to a certain extent [26]. EVs derived from MSCs have more advantages than that of MSCs. First, the phospholipid bilayer vesicles of EVs prevent themselves from being recognized as foreign objects by tissues and become complex carriers that protect enzymes, cytokines, and genetic material from degradation. Moreover, owing to the presence of cell-binding affinity proteins embedded on the surface of vesicles, EVs can show the same excellent delivery efficiency as MSCs. Second, EVs move freely in the blood because of their nanometer size, which can easily achieve membrane fusion of target cells and can penetrate the skin mucosal barrier, blood–brain barrier, and placental barrier, making them an ideal carrier for the delivery of active molecules and drugs [27]. However, the ratio of MSCs enriched to the target site through blood circulation after administration and the ratio of MSCs integrated to the damaged site in a short time are relatively low. Third, EVs are less immunogenic than MSCs because they do not express MHC-I or MHC-II antigens on their membrane surfaces; thus, tissues do not recognize these EVs as foreign, protecting their contents from degradation. However, MSCs express high levels of MHC-II when stimulated by inflammation, and treatment with MSCs was reported to be carcinogenic [28]. Fourth, EVs are highly modifiable as noncellular structures. By loading functional drugs, specific proteins, and non-coding RNAs, including miRNAs and siRNAs, EVs can replace cell therapy and become a new biotherapeutic method.

## 3. Mechanism of MSC-EVs in Promoting Wound Healing and Skin Regeneration

Skin wound healing is a series of physiological processes that begins after the normal anatomical structure or integrity of the skin is destroyed. Several studies have investigated the effects of MSC-EVs on wound healing and skin diseases. Different types of cells are involved in different stages of chronic wound healing (e.g., hemostasis, inflammation, proliferation, and remodeling), including immune cells involved in the regulation of inflammation, for example, macrophages and neutrophils T cells [29], and cells involved in tissue proliferation and remodeling, for example, fibroblasts, keratinocytes, and endothelial cells (Figure 1). In this chapter, we attempt to elucidate the immunomodulatory effects of MSC-EVs on different cell types.

### 3.1. MSC-EVs Regulate Neutrophil Changes

Neutrophils are the first line of defense against acute inflammation. During the inflammatory phase of skin injury, neutrophils first infiltrate the site of injury to sweep microbial pathogens and then undergo apoptosis, after which macrophages infiltrate and engulf phagocytic fragments, apoptotic neutrophils, and other apoptotic cells. Studies have shown that keratinocyte-derived EVs treated with cytokines can significantly induce neutrophil production and release and can induce the expression of IL-6, IL-8, and tumor necrosis factor-*α* (TNF-α) in neutrophils through the nuclear factor kappa B (NF-κB) and P38 MAPK signaling pathways [30]. However, prolonged inflammatory processes can lead to increased fibroblast activity and extracellular matrix oversecretion, leading to skin scarring initiated by lymphocytes and hypertrophic scarring caused by multiple directions or intermittent strain of the scar [31]. Therefore, scar reduction focuses on modulating neutrophil changes, stimulating scar maturation, and reducing the inflammatory response and myofibroblast production. Complement and neutrophils are two key elements of the innate immune system [32]. Evidence suggests that MSC-EVs inhibit complement activation via CD59, disrupting the feedforward loop between complement and neutrophils and inhibiting the amplification and persistence of inflammation during infection [32]. Furthermore, topical application of MSC-EVs inhibits the activation of cuticle complement, alleviating the release of IL-17 by neutrophil extracellular traps that accumulate in and below the cuticle [33]. Additionally, MSC-EVs have been shown to affect neutrophil-mediated microvascular remodeling [34].

### 3.2. MSC-EVs Regulate Macrophage Changes

Macrophages are derived from monocytes, participate in non-specific defense (innate immunity) and specific defense (cellular immunity) in vivo, and can differentiate into two activated subtypes, M1 and M2. M1 macrophages are typically activated cells that secrete many pro-inflammatory factors, such as TNF-α, IL-1β, and reactive oxygen species; M2 macrophages alternately activate and produce anti-inflammatory cells that produce IL-10 and trophic factors. Ko reported that the enhancement of macrophage phagocytic activity by MSCs depended on the uptake of EVs containing MSC mitochondria [35]. Accumulating evidence suggests that MSC-EVs play a relevant role in regulating M1/M2 equilibrium, although the exact mechanism remains unclear [36]. Evidence suggests that MSCs release the C–C chemokine ligand (CCL2, or monocyte-chemoattractant protein, MCP-1), recruiting monocytes/macrophages to the injury site and supporting wound healing [37]. MSC-EVs have been reported to possibly hinder the activation of “pro-inflammatory” M1 macrophages in favor of “pro-lytic” M2 macrophages [38]. For example, treatment with MSC-EVs can significantly enhance the gene expression of M2 macrophage markers (Arg1, CCL22, IL-10) and transforming growth factor-β (TGF-β) [39]. MSC-EVs also downregulate IL-23 and IL-22 production and enhance the anti-inflammatory phenotype and pro-decomposition properties of Mregs (human regulatory macrophages, a subclass of M2 macrophages characterized by moderate IL-22 and IL-23 production and high prostaglandin E2 expression) [40]. M2 macrophages secrete anti-inflammatory cytokines and various growth factors that play an important role in wound healing, and modulation of M2 polarization through MSC-EVs can promote skin wound healing. In addition, treatment with MSC-EVs attenuated the score and degree of skin fibrosis in the scleroderma chronic graft versus host disease (cGVHD) mice model, possibly by decreasing the percentage of macrophages in the skin and spleen, reducing the infiltration of macrophages into the skin, and reducing the production of TGF-β and Smad2 in the skin [41]. EVs derived from human umbilical cord MSCs (hUC-MSCs) can mediate TLR4 signaling through miR-181c, reducing the inflammatory response of macrophages in burned rats [42]. Stem cells from human exfoliated deciduous teeth (SHED) can generate EVs to stimulate macrophage autophagy, reduce itching, and promote inflammatory wound healing [43]. In promoting wound healing in diabetic skin, MSC-EVs have also been shown to reduce inflammation by enhancing macrophage polarization [44]. In a lipopolysaccharide-induced wound model, SHED-EVs enhanced the autophagy function of macrophages through the AKT, ERK1/2, and STAT3 signaling pathways to promote wound healing, reducing itching [45]. Melatonin-pretreated MSC-EVs significantly inhibited the pro-inflammatory cytokines IL-1β and TNF-α, activated the PTEN/AKT signaling pathway, increased the ratio of M2 to M1 polarization, and inhibited the inflammatory response, promoting diabetic wound healing [45]. In addition, keratinocyte-derived EVs are considered major contributors to the regulation of macrophage trafficking and maintenance of the epithelial barrier after injury [46].

### 3.3. MSC-EVs Regulate T Cell Changes

T cells are derived from bone marrow pluripotent stem cells and are involved in many aspects of adaptive immunity [47]. Several observations suggest that T cells play an important role in the regulation of inflammation in skin repair [48]. An effective method to suppress T cell-mediated regulation of inflammation is to prevent T cell proliferation. MSCs have immunomodulatory properties that induce suppressive T cell production. The immunosuppressive function of MSCs is enhanced by interferon-γ (IFN-γ) and TNF-α produced by T cells, which can be further amplified by cytokines, such as IL-17 [49]. MSC-EVs have also been found to play critical roles in many T cell-mediated reactive conditions and have received increasing attention as immunomodulators and anti-inflammatory agents. Evidence suggests that MSC-EVs can switch activated T cells to a T-regulatory phenotype, suppressing inflammatory responses [50]. In several studies, MSC-EVs exerted this effect in several animal models, in vitro and in vivo. For example, in vivo injection of MSC-EVs can significantly suppress the immune response of cytotoxic T cells (Tc1 cells) and type 1 helper T cells (Th1), reduce pro-inflammatory TNF-α and IFN-γ levels, and induce regulatory T cells (Tregs) and anti-inflammatory IL-10 levels, preventing the onset of allergic contact dermatitis (ACD, a typical T cell-mediated disease) in a mouse model [51]. This process was further validated by in vitro experiments, and MSC-EVs were found to alter the metabolism of Th1-type differentiated T cells in relation to the TGF-β pathway [52]. In Su’s in vivo experiments, immune cells dominated the uptake of MSC-EVs from the biofunctional scaffolds. Scaffolds and exosomes act as recruiters and trainers of immune cells, respectively, synergistically promoting Treg responses in mouse skin trauma [53].

### 3.4. MSC-EVs Regulate Fibroblast, Keratinocyte, and Endothelial Cell Changes

Four main regeneration phases occur during hyperplasia and remodeling: fibroblast proliferation, extracellular matrix (ECM) component production, re-epithelialization, and angiogenesis [54]. After an inflammatory phase involving a multi-cytokine burst, the damaged site begins to regenerate new tissue to restore skin form and function [54]. A large body of evidence suggests that MSC-EVs exert a positive therapeutic effect on these four processes. For example, fibroblasts absorbed MSC-EVs and showed a significant dose-dependent increase in cell proliferation and migration [55]. MSC-EVs can also promote and optimize collagen deposition (e.g., type I collagen and type III collagen) in vitro and in vivo and further promote wound healing through the PI3K/AKT signaling pathway [55,56]. Simultaneously, EVs derived from adipose-derived mesenchymal stem cells (AD-MSCs) can regulate the proportion of collagen type III and I, TGF-β3, TGF-β1, matrix metalloprotease (MMP) 3, and TIMP1 and can reduce scar formation by regulating fibroblast differentiation during skin wound repair [57]. Furthermore, human amniotic epithelial cell (hAECs)-derived EVs have been shown to accelerate wound healing by promoting fibroblast proliferation and migration [58,59]. By contrast, many researchers use keratinocytes exposed to hydrogen peroxide (H2O2) to establish a skin injury model and treat the model with MSC-EVs to observe its efficacy. AD-MSC-derived EVs were found to promote the proliferation and migration of skin injury model cells, reduce apoptosis, and play an active role in skin wound healing through the Wnt/β-catenin signaling pathway [60]. EVs derived from bone marrow MSCs (BM-MSCs) have also been shown to promote the proliferation and migration of skin injury model cells, inhibit apoptosis, and accelerate wound healing through the Mir-93-3p/APAF1 axis [61]. Re-epithelialization, collagen deposition, and neovascularization are successive, inseparable processes, and many researchers have discussed multiple cell types simultaneously while exploring the promotion of MSC-EVS for injury repair. For example, Shabbir showed that MSC-EVs enhance fibroblast proliferation and migration and increase human umbilical vein endothelial cell tubulogenesis in a dose-dependent manner [62]. Furthermore, MSC-EVs enhance fibroblast migration by activating intracellular pathways such as AKT, ERK, and STAT3, which are all vital pathways in wound healing [62]. Ren investigated the effects of AD-MSCs on fibroblasts, keratinocytes, and endothelial cells and observed changes in vivo [63]. The results showed that MSC-EVs significantly promoted the proliferation and migration of these cells through the AKT and ERK pathways; upregulated growth factors, such as VEGFA, PDGFA, EGF, and FGF2; and enhanced re-epithelialization, collagen deposition, and neovascularization in in vivo skin injury models, accelerating wound healing [63].

Wound healing is a multi-step process involving complex pathways at the cellular and molecular levels. Identifying functional gene variants, such as single nucleotide polymorphisms (SNPs), that are closely associated with wound healing and the establishment of venous ulcers will greatly assist in the prognosis, diagnosis, and treatment of chronic wounds [64]. MSC-EVs regulate cell differentiation and proliferation, affect angiogenesis, interfere with stress responses, and participate in immune signaling [65]. A systematic assessment of 66 patients treated with adipose-derived stem cells (ADSCs) for venous leg ulcer (VLU) showed significantly higher wound-healing rates compared to controls, decreased pain scores, and no serious surgery-related complications reported [66]. Thus, it is speculated that MSCs and MSC-EVs may also contribute to the prognosis, diagnosis, and treatment of chronic wounds by influencing the role of SNPs.

Therefore, EVs derived from various MSCs (e.g., AD-MSCs, BM-MSCs, hUC-MSCs, hAECs, etc.) have substantial therapeutic effects on skin wound healing by reducing the inflammatory response, promoting re-epithelialization and angiogenesis, promoting the proliferation and migration of fibroblasts, and enhancing the formation and remodeling of ECM.

### 3.5. MSC-EVs Promote Scar-Free Repair of Skin Wounds

After the dermal tissue is damaged, the abnormal deposition of collagen in the extracellular matrix and the accumulation of fibroblasts often cause scar hyperplasia, which is characterized by thickening, hardening, redness, and itching of the scar, affecting the quality of life of patients [67]. This mechanism is related to the abnormal immune function of T cells and macrophages. TGF-β1 stimulates the proliferation of collagen fibers in wound tissue and inhibits the decomposition of ECM, such as collagen, by MMPs, promoting scars [68]. Additionally, if the wound surface has an insufficient blood supply, delayed wound healing aggravates scar hyperplasia [69]. Traditional skin damage repair methods have certain curative effects, which can reduce patients’ pain and promote wound healing; however, the clinical curative effect is limited, the long-term curative effect is not ideal, and the problem of scar tissue hyperplasia cannot be solved [69]. Studies have confirmed that AD-MSCs can promote wound healing and scar formation; however, direct transplantation of AD-MSCs leads to a low survival rate and limited clinical efficacy [1]. Nevertheless, EVs generated by AD-MSCs can be cell-free and help promote cell damage repair, which effectively accelerates the repair of skin wounds and reduces the generation of scars, helping to improve the beauty of skin healing [57,70]. The EVs produced by AD-MSCs help promote the healing of skin wounds and simultaneously reduce the accumulation of fibroblasts to a certain extent, reducing the generation of scars and improving the quality of skin wound healing. The mechanism of action of AD-MSC-EVs is complex and can activate various signaling pathways and promote the release of wound-related factors. The formation of wound scars is often accompanied by the deposition, degeneration, and degradation of the cell matrix, and cell matrix proteins are the main substances that degrade the cell matrix. Studies have shown that keloid cells express high levels of MMP1 and MMP3, which can promote scar formation by regulating the cell matrix, whereas AD-MSC-EVs can inhibit the expression of MMP1 and MMP3 to a certain extent and reduce the degradation of the peripheral cytoplasmic matrix, reducing scar formation [71].

In summary, MSC-EVs can promote the scar-free repair of the skin by regulating the remodeling of the extracellular matrix, increasing the expression of matrix metalloproteinases, promoting the reconstruction of the extracellular matrix, effectively inhibiting the formation of scars, improving the quality of skin repair, and improving the quality of skin repair. The treatment of clinical skin wound healing plays a guiding role.

### 3.6. Application of MSC-EVs in Animal Studies

Recent studies on MSC-EVs have focused mostly on in vivo experiments for the treatment of diseases with complex pathophysiology. For example, through in vivo evaluation of BM-MSC-EVs in the treatment of corneal epithelial wounds, it was found that in the corneal injury mouse model, MSC-EVs effectively promoted wound healing, significantly reduced haze/edema of the cornea after treatment, and decreased corneal fibrosis markers, fibronectin, collagen 3A1 and a-SMA [72,73]. In a rat model, local application of MSC-EVs can significantly accelerate the mucosal healing of oral mucositis (OM) [74]. In Levy’s data, the pro-angiogenesis and anti-inflammatory activity of IPSC-derived MSC-EVs has been identified in a mouse model of diabetic wound healing and effectively mediates inflammation resolution within the wound bed [75]. Interestingly, the EVs secreted by un-induced iPSCs increased the rate of new epithelial formation at earlier time points and decreased the total wound area and length at later time points [75]. At the same time, compared with BM-MSC-EVs, EVs derived from oral mucosa Lamina propria progenitor cells (OMLP-PC) is more effective in driving wound repair and scar healing [76]. This may imply that the biological effects of EVs are at least partially influenced by the pluripotency of the source cells.

In addition, given the characteristic of MSC-EVs promoting wound healing, some bioengineering materials have been applied to extend their therapeutic efficacy. For example, thermosensitive chitosan-based hydrogel (CHI hydrogel) sustained-release iPSC-MSC-EVs can effectively promote the repair of damaged corneal epithelium and stroma, downregulate collagen mRNA expression, and reduce scar formation in vivo [77]. Using the biotin–avidin interaction, biotin-modified MSC-EVs were fixed in avidin-linked GelMA to produce GelMA-EVs hydrogels, which increased the slow-release effect of EVs, and MSC-EVs were observed to remain more stable in GelMA-EVs hydrogels for up to 28 days after subcutaneously transplanted [78]. This in vivo evidence suggests that MSC-EVs have been shown to promote wound healing and reduce scar area in animals, whether administered directly or after optimized biological characteristics (such as increasing the pluripotent capacity of source cells, combining biomaterials, etc.).

## 4. Challenges in Applying MSC-EVs to Promote Wound Healing and Skin Regeneration

Cell therapy has made great strides in the clinical practice of skin damage repair, and an increasing number of clinical trials have reported the therapeutic effects of MSC-EVs. As a new therapeutic approach, MSC-EVs have many limitations that must be overcome before they can be used clinically. First, their effects are difficult to predict in vivo because of the tissue origin, concentration, number of doses, route and timing of MSC administration, and inflammatory state of the recipient. To predict the biological effects of MSC-EVs, a comprehensive characterization of MSC-EV content and standardization of experimental methods are essential. The MSCs donors used to generate EVs need to be planned and regulated, and a standard good manufacturing practice (GMP)-compliant MSC-EV isolation protocol needs to be developed and refined. Second, because MSC-EVs cover a relatively wide range involving microvesicles (MVs), apoptotic bodies, and exosomes, developers need to classify MSC-EVs and establish consistent, graded release criteria (e.g., particle size, loading, surface marker expression) before they can be injected into potential patients. Additionally, medical practitioners need to monitor the treatment process in the human body at any time, determine the markers that distinguish functional and non-functional EVs based on the efficacy of the treatment, and then report them back to researchers to pursue the production of functional-specific MSC-EVs. Third, the optimal dose of MSC-EVs in humans, the optimal route of administration of MSC-EVs, and the length of time that MSC-EVs remain in patients before being cleared by phagocytes remain unclear and need to be determined according to the treatment. Investigators must overcome these limitations to achieve MSC-EV-induced immunomodulation and regeneration.

## 5. Conclusions

Poor skin wound healing is a common problem in the field of trauma medicine. The different stages of chronic wound healing (hemostasis, inflammation, proliferation, and remodeling) require the involvement of different types of cells, including immune cells involved in the regulation of inflammation, such as macrophages, neutrophils, and T cells, and cells involved in tissue proliferation and remodeling, such as fibroblasts, keratinocytes, and endothelial cells. MSC-EVs have been shown to facilitate skin wound healing and accelerate this process through multiple mechanisms. These mechanisms comprise reducing inflammation, promoting angiogenesis, promoting the proliferation and migration of epithelial cells and fibroblasts, and ECM remodeling. Overall, MSC-EVs, an important component of the ECM, play a crucial role in skin injury and are closely related to tissue regeneration. Therefore, as new biomarkers and therapeutic targets, MSC-EVs can simultaneously promote wound healing through multiple mechanisms and are a promising approach to replacing cells for skin wound therapy.

## Figures and Tables

**Figure 1 genes-14-01516-f001:**
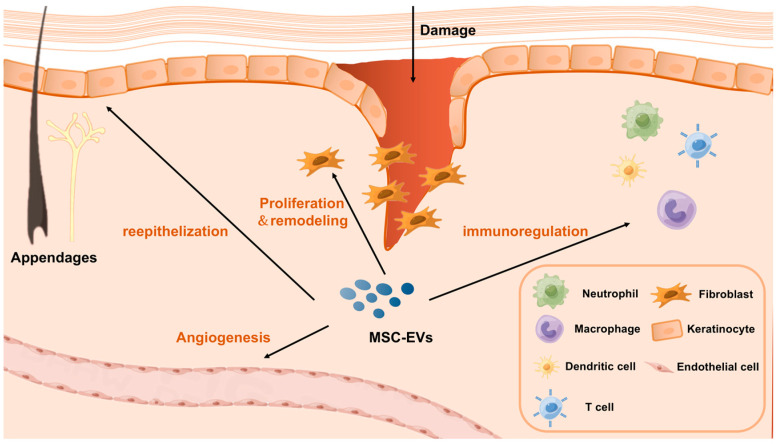
A schematic diagram illustrating the mechanism of MSC-EVs in promoting wound healing and skin regeneration.

## Data Availability

Not applicable.

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
