# Peer review of "Extracellular Vesicles Derived from Mesenchymal Stem Cells Promote Wound Healing and Skin Regeneration by Modulating Multiple Cellular Changes: A Brief Review"

_genes, 2023, doi:10.3390/genes14081516_

Round 1

Reviewer 1 Report

The paper of Zhang et al. is reviewing the properties of EV derived from mesenchymal stem cells in wound healing. The data presented are interesting although the reference consulted are not completely updated, most of the papers stop at 2020. I have few questions to the authors:

line 28: what are the “chemical molecular drugs” the authors claim are used in wound dressing?

Line 46: what is the nutritional effect? Please explain

Line 67: MSC are not defined as “an adherent state”

Line 73: delete the phrase please

Line 92: please delete scholars, also in other phrases

Line 95: please do not repeat that EV contains proteins, nucleic acids and lipids

Line 177: the pro-decomposition properties of Mregs are not clear to me, please explain

The conclusions are almost the same as the introduction. Please update the references

Author Response

Thank you for your reply. In response to your valuable questions, our explanations are as follows:

line 28: Changed to "systemic anti-inflammatory drugs”.

line 46: phrase "nutritional effect" has been removed.

line 67: In fact MSCs are not restricted to an adherent state, the original sentence has been modified.

line 73: Duplicates have been removed.

line 92: Repeated statements have been modified.

line 95: Duplicates have been removed.

line 177: Mregs are actually defined as human regulatory macrophages, a subclass of M2 macrophages characterized by moderate IL-22 and IL-23 production and high prostaglandin E2.

The conclusions have been modified, and references have been updated. please review the updated version.

Reviewer 2 Report

Dear Authors: 

The paper presented for review concerns a review of publications on the important issue of wound healing. Difficult-to-heal wounds are a very important problem of modern medicine, which is closely related to the problem of civilization diseases such as type II diabetes mellitus, or the increasing resistance of small-breeds. Of course, the use of microbubbles in therapy does not always solve these problems, but can be an important step in this direction. After reading the paper, I have the following questions and comments?

1. Why the short review, the topic is important and the work could have been more elaborated.

2. For chapter 3 should be based on diagrams showing possible mechanisms of action. 

3. Please add a summary of clinical trials and animal studies.

Author Response

Thank you for your valuable letter. In response to your questions, our explanations are as follows:

1. The review has been expanded and updated.

2. A detailed elaboration of MSC-EVs for the regulation of the immune system is necessary. To this end, we have added a sub-section for supplementary explanations.

3. Relevant reviews of in vivo trials are critical and relevant chapters have been added.

Thank you again for your letter.

Round 2

Reviewer 1 Report

The authors complied with the requests.